# From Mechanism to Therapy: The Role of MSC-EVs in Alleviating Radiation-Induced Injuries

**DOI:** 10.3390/pharmaceutics17050652

**Published:** 2025-05-16

**Authors:** Chong Huang, Heng Li, Zhiyue Zhang, Ting Mou, Dandan Wang, Chenlu Li, Lei Tian, Chunlin Zong

**Affiliations:** 1State Key Laboratory of Oral & Maxillofacial Reconstruction and Regeneration, National Clinical Research Center for Oral Diseases, Shaanxi Key Laboratory of Stomatology, Department of Oral and Maxillofacial Surgery, School of Stomatology, The Fourth Military Medical University, Xi’an 710032, China; huangchongson@163.com (C.H.); 17798815711@163.com (H.L.); zhangzy126830@163.com (Z.Z.); 17791823751@163.com (D.W.); lcl17309283208@outlook.com (C.L.); 2School of Stomatology, Jiamusi University, Jiamusi 154007, China; mtmuting@126.com

**Keywords:** mesenchymal stem cells, extracellular vesicles, radiation-induced injuries

## Abstract

Radiation injury is a severe issue in both nuclear accidents and cancer radiotherapy. Ionizing radiation impairs the regenerative and repair capabilities of tissues and organs, resulting in a scarcity of effective therapeutic approaches to prevent or mitigate such injuries. Mesenchymal stem cells (MSCs) possess favorable biological characteristics and have emerged as ideal candidates for the treatment of radiation injury. However, the use of MSCs as therapeutic agents is associated with uncertainties in therapeutic efficacy, transient effects, and the risk of immune rejection. Recent advances in research have revealed that extracellular vesicles (EVs) derived from mesenchymal stem cells (MSC-EVs) exhibit similar beneficial properties to MSCs and represent a promising cell-free therapy for mitigating radiation injuries. MSC-EVs are enriched with microRNAs (miRNAs), proteins, and lipids, which can modulate immune responses, inflammatory reactions, cell survival, and proliferation in irradiated tissues. This review synthesizes recent studies on the application of MSC-EVs in radiation injury, focusing on the therapeutic effects and mechanisms of MSC-EVs derived from various sources in radiation-induced diseases of different organs. The therapeutic potential of MSC-EVs for radiation injury provides valuable insights for addressing ionizing radiation-induced injuries and offers a reference for future clinical applications.

## 1. Introduction

Nuclear accidents and cancer radiotherapy are two major sources of ionizing radiation injury in humans [1]. During nuclear accidents, along with the large-scale, uncontrolled release of radioactive substances, surrounding populations were subject to intense and extensive short-term ionizing radiation impacts [2]. In the short term, high-dose ionizing radiation can trigger acute radiation sickness, with symptoms like nausea, vomiting, and skin erythema. Severe cases may lead to bone marrow suppression, immune system injury, and could even be life-threatening [3,4]. Radiotherapy, a crucial cancer treatment, precisely targets cancer cells but inevitably exposes surrounding healthy tissues to some radiation [5]. Radiation-induced injury from radiotherapy can also cause severe harm, such as bone marrow suppression, intestinal inflammation, skin ulcers, radiation-induced pulmonary fibrosis, radiation-induced bone necrosis, and so on [6,7,8]. Ionizing radiation directly damages biomacromolecules such as DNA, proteins, and lipids by transferring energy. Additionally, when radiation rays traverse cells, it induces the ionization of water molecules, generating reactive oxygen species (ROS) such as superoxide anions and hydroxyl radicals. The ROS can oxidize DNA molecules, leading to base damage and single-strand and double-strand breaks, which finally result in cell cycle arrest, cell senescence, and cell death.

Current radiation countermeasures mitigate radiation-induced damage through multiple approaches, such as antioxidant properties, immune modulation, tissue repair and regeneration promotion, anti-inflammatory and anti-fibrotic therapies, and the regulation of oxidative stress and apoptosis [9]. Several agents have been identified to alleviate radiation damage, including cytokines (e.g., granulocyte colony-stimulating factor (G-CSF), granulocyte–macrophage colony-stimulating factor (GM-CSF), and erythropoietin (EPO)), thrombopoietin receptor agonists (e.g., Nplate, Eltrombopag, and Lusutrombopag), and pegylated drugs (e.g., Neulasta, Pegasys, and Sylatron). These agents have received approval from the U.S. Food and Drug Administration (FDA) for the treatment of acute high-dose radiation exposure. However, their efficacy is limited, and their considerable toxicity and side effects pose challenges to achieving satisfactory clinical outcomes [10,11,12,13].

Mesenchymal stem cells (MSCs) are multipotent cells characterized by high proliferation and differentiation potentials [14]. Previous studies have shown that MSCs could affect physiological processes such as immune regulation, hematopoiesis, tissue repair, regeneration, and anti-apoptosis. Studies indicated that extracellular vesicles (EVs) from MSCs (MSC-EVs) share functional similarities with their parent cells. MSC-EVs possess stem cell-like repair capabilities for injured tissues in blood, heart, lungs, bone marrow, intestines, and skin. EVs were initially considered as a metabolic waste export pathway. However, EVs release contents like miRNA and proteins that mitigate tissue injury via anti-apoptosis, immune modulation, anti-inflammation, anti-fibrosis, and angiogenesis, showing great potential in radiation injury prevention and treatment [15,16,17]. Therefore, researching the therapeutic effects and mechanisms of MSC-EVs in radiation injury repair is essential for developing new treatment strategies.

This review explores the therapeutic effects and mechanisms of MSC-EVs from various sources, including bone marrow, adipose tissue, umbilical cord, placenta, and dental pulp, in radiation injury repair. It summarizes their role in treating radiation injuries of different organs, such as bone, lung, intestines, liver, skin, heart, and brain. Here, we also discuss the potential limitations and challenges associated with the clinical translation of MSC-EVs, providing new insights for treatment research in radiation injury.

## 2. Pathophysiology and Current Therapeutic Landscape of Radiation Injury

### 2.1. Pathophysiology of Radiation Injury

Radiation injury is a multidimensional biological destruction induced by ionizing radiation, manifesting not only as acute tissue necrosis and organ dysfunction but also posing long-term threats such as genetic mutations and carcinogenic risks [18]. The mechanisms of radiation damage are shared among various cells, although tissue-specific symptoms and pathological features are exhibited. At the molecular level, radiation directly induces DNA and protein, resulting in destabilization of genetic material [19]. Concurrently, it triggers lipid peroxidation and functional disruption of membrane proteins, leading to abnormal membrane permeability [20]. Moreover, mitochondria, as radiation-sensitive targets, suffer from DNA injury, collapse of membrane potential, and bursts of reactive oxygen, resulting in exacerbated oxidative stress, disrupted energy metabolism, and activation of apoptotic signaling [21]. Furthermore, radiation causes oxidative protein modifications and RNA injury, leading to a compromised cellular functional homeostasis. Cells initiate G1/G2 phase arrest via the ATM/ATR-Chk1/2 pathway for injury repair; however, severe injury activates p53-dependent apoptosis, resulting in proliferative inhibition and differentiation abnormalities. Metabolic dysregulation manifests as impaired ATP synthesis and redox imbalance, while the immune system amplifies inflammatory responses through pro-inflammatory cytokines and immune cell infiltration [22]. These multidimensional injuries culminate in irreversible genetic injury, such as mutations and chromosomal aberrations, forming the core pathological network of radiation injury.

The pathological manifestations of radiation injury exhibit marked tissue specificity, with severity closely linked to cellular proliferation kinetics, radiosensitivity, and repair capacity of organs. Highly proliferative tissues (e.g., bone marrow and intestinal epithelium) are more susceptible to radiation-induced DNA breaks due to active DNA replication, whereas terminally differentiated tissues (e.g., neural and muscular tissues) are prone to delayed injuries owing to limited repair capacity. Additionally, microenvironmental features (e.g., vascular density and oxidative stress levels) further modulate the spatiotemporal evolution of radiation injury [23]. We have summarized the common organs in the human body that are susceptible to radiation damage, as well as the impact of macromolecular damage on cellular function during radiation injury (Figure 1).

### 2.2. Current Treatment of Radiation-Induced Diseases

There was little difference in therapeutic purposes, but the treatment measures of radiation-induced diseases in different organs varied because of the anatomic difference. The current treatment for radiation injury in bone, intestines, pulmonary, skin, and brain are listed hereinafter.

Radiation-induced bone injury refers to bone tissue damage caused by radiation therapy, including bone marrow hematopoietic suppression, bone structure destruction, osteonecrosis, pathological fractures, etc. Hyperbaric oxygen therapy (HBOT) is a method used to promote tissue healing and is used as an adjunct to surgical and antimicrobial treatments. The Marx protocol is widely used by many doctors; it involves receiving 30 HBOTs before surgery, followed by surgery, and receiving 20–30 treatments after surgery [24]. However, an increasing number of studies suggest that HBO has limited effectiveness and its effectiveness cannot be proven. For example, Nolen et al.’s research shows that the incidence of infection complications increases in flap transplant patients receiving HBOT [25]. Surgical removal of dead bone and diseased tissue is one of the routine treatment methods for radiation-induced bone injuries. Surgical intervention is a necessary choice for patients who have failed conservative treatment. However, due to impaired blood supply after radiation, postoperative bone healing ability is poor, leading to a high risk of surgical failure and complications [26,27]. At present, there is no specific drug that can completely reverse the pathological process of radiation-induced bone injury. Antifibrotic drugs can inhibit the development of bone fibrosis and improve the microenvironment of bone tissue [28]. Antibiotics are mainly used to control infections and prevent complications such as osteomyelitis. In addition, studies have found that some drugs, such as amifostine and lichen acid, had certain potential in preventing and treating radiation-induced bone damage [29]. Stem cell therapy has brought new hope for the repair of radiation-induced bone injuries. Various types of stem cells, such as bone marrow mesenchymal stem cells, adipose-derived mesenchymal stem cells, and umbilical cord blood mesenchymal stem cells, are used to treat radiation-induced bone injury [30]. These stem cells have the potential to differentiate into osteoblasts, which can promote the regeneration and repair of bone tissue.

Mild radiation enteritis can be treated with symptomatic treatments such as montmorillonite powder and probiotics. For moderate to severe patients, medications such as mesalazine can be added. For chronic radiation enteritis, research has explored the use of immunosuppressive agents such as thioguanine and stem cell therapy [31]. In addition, there are also traditional Chinese medicine treatment methods such as Anal Tai suppository and Kangfuxin liquid being used in clinical practice [32]. The efficacy of existing therapeutic drugs for chronic radiation enteritis is limited and cannot fundamentally reverse intestinal damage. Long-term use of immunosuppressants may have side effects, and new methods such as stem cell therapy are still in the research stage and have yet not been widely applied [33].

Radiotherapy for thoracic tumors can sometimes cause radiation pneumonitis or even pulmonary fibrosis [34]. Traditional therapy still mainly relies on glucocorticoids and symptomatic treatment. Although it can alleviate some symptoms, its effect on improving lung diffusion and ventilation function is not ideal [35]. In recent years, clinical trials of some new drugs, such as pirfenidone, which is an anti-fibrotic medicine, have achieved positive results with few side effects, i.e., improvements in lung function, mild-to-moderate radiation-induced pulmonary fibrosis, and survival rate without acute lung deterioration in patients with radiation-induced lung injury [36]. At present, there is still a lack of drugs that can clearly prevent radiation-induced pulmonary fibrosis. For patients with pulmonary fibrosis, treatment methods are limited and ineffective. Long-term efficacy and safety still need to be further observed and verified. The existing treatment guidelines have a low recommended level and lack high-quality clinical research evidence to support it.

For radiation-induced skin injury, there are many treatment methods, such as oral and inhalation drug treatment (Epigallocatechin gallate, N-Acetyl-L-cysteine, etc.), local drug treatment (corticosteroids, Atorvastatin gel, traditional Chinese medicine ulcer oil, etc.), pulsed dye laser treatment, photobiomodulation therapy, HBOT, and so on [37]. All these methods were aimed at relieving symptoms, inhibiting inflammation, and promoting skin regeneration. However, they are not effective for chronic and severe radiation-induced skin injury, and surgical treatment is needed. Therefore, new treatment research has also drawn more attention to MSCs, metformin, and hydrogen therapy [38].

For radiation-induced brain injury, methods including drugs, surgery, HBOT, and rehabilitation were used [39]. Drugs mainly consist of glucocorticoids, anti-angiogenic drugs, antioxidants, antiepileptics, and nootropics. Glucocorticoids, such as dexamethasone and prednisone, are commonly used in clinical practice to reduce brain edema and inflammation. Anti-angiogenic drugs, such as Bevacizumab and Apatinib, can reduce vascular permeability and inhibit abnormal angiogenesis. Antioxidants, such as melatonin, vitamin E, and Edaravone, can clear free radicals, reduce oxidative stress damage, and improve brain metabolism. Antiepileptics, such as sodium valproate, can stabilize nerve cell membranes and resist inflammation. Olacetam, as a nootropic, can promote brain metabolism and improve cognitive function. However, all these measures only alleviate symptoms and improve brain function to a certain extent, which cannot reverse the damage and bring a full recovery [40].

In summary, the treatment strategies for radiation-induced diseases cover various means, including drug therapy, surgical treatment, HBOT, rehabilitation therapy, stem cell therapy, etc. However, there are many limitations, such as short-term or uncertain drug efficacy, large surgical trauma, high risk, lack of large-scale experimental support for HBOT, and immature stem cell technology.

## 3. Biological Characteristics of MSC-EVs

The term “EVs” herein collectively refers to exosomes (30–100 nm), microvesicles (100–1000 nm), and apoptotic bodies (500–4000 nm). Exosomes derived from mesenchymal stem cells first form intracellular vesicles (ILVs) through transport proteins and then fuse with the plasma membrane before secretion [41]. Microvesicles and apoptotic bodies are directly released from the plasma membrane. EV secretion involves one or more mechanisms, including the endosomal sorting complex required for transport (ESCRT) complex, tetraspanins, ceramide-generating sphingomyelinases, phospholipid redistribution to budding sites, and actin cytoskeleton depolymerization [42]. MSC-EVs possess a range of distinctive biological characteristics. In terms of physical properties, they are typically nanoscale membrane vesicles with a bilayer lipid structure, exhibiting various morphologies such as cup-shaped or round. Functionally, MSC-EVs have significant paracrine effects. They can participate in tissue repair and regeneration by promoting cell proliferation, differentiation, and angiogenesis. For instance, in injured tissues, they deliver bioactive molecules to stimulate resident cells to repair damage and restore tissue function [43,44].

EVs serve both as a means for secreting cells to dispose of harmful or redundant components and as critical mediators of intercellular communication. They carry diverse bioactive cargoes capable of transmitting signals and inducing physiological changes in recipient cells. EV biogenesis and release are regulated by a cellular metabolic status, microenvironmental cues (e.g., hypoxia and inflammatory cytokines), and mechanical stimuli, resulting in marked heterogeneity in EV composition and function across sources and conditions [45]. Studies indicate that radiation enhances cellular EV uptake and alters EV composition/secretion. Hypoxia may augment the pro-angiogenic effects of MSC-EVs via HIF-1α-mediated activation of miR-210 and neutral sphingomyelinase 2 [46,47]. This spatiotemporal regulation of secretion underscores the functional plasticity of EVs, providing a theoretical foundation for their use in radiation injury therapy.

Previous studies highlight the therapeutic potential of MSCs in repairing radiation-injured tissues. MSC-EVs deliver critical growth factors and signaling molecules via paracrine mechanisms [48,49]. Unlike fragile live cells, MSC-EVs retain bioactivity under repeated manipulations due to their lipid bilayer structure. MSC-EVs encapsulate DNA, miRNAs, mRNAs, proteins, and lipids, which mediate intercellular signaling. For radiation injury, a key mechanism involves transmembrane delivery of functional molecules. For instance, miR-21-5p overexpression in MSC-EVs promotes angiogenesis by upregulating VEGF, Arg-1, and Tie-2 [50].

EVs exhibit intrinsic tropism to injured tissues, a prerequisite for their therapeutic efficacy. Integrin β1 and α4 interactions are critical for EVs uptake and RNA delivery [51]. Engineered EVs can be loaded with diverse therapeutic cargoes (e.g., siRNA, chemotherapeutics, and immunomodulators) and targeted to specific sites [52]. Their biocompatibility minimizes uptake by the reticuloendothelial system, prolonging circulation and protecting payloads until delivery.

## 4. Therapeutic Effects and Potential Mechanisms of MSC-EVs in Radiation Injury

In the research of radiation treatment by MSC-EVs, the main sources of MSCs are bone marrow, adipose, placenta, umbilical cord, and dental pulp (Figure 2). Therefore, the MSC-EVs were named, respectively, as bone marrow mesenchymal stem cell-derived EVs (BMSC-EVs), adipose-derived stem cell-derived EVs (ADSC-EVs), placental mesenchymal stem cell-derived EVs (PL-MSC-EVs), umbilical cord mesenchymal stem cell-derived EVs (UCMSC-EVs), and dental pulp mesenchymal stem cell-derived EVs (DPSC-EVs). Different resources give the MSCs unique features; for instance, BMSC is the first discovered and widely used MSC; ADSC and DPSC are easier and safer to obtain; and PL-MSC and UCMSC have stronger clone formation and differentiation abilities. MSC-EVs from multiple sources were used to treat radiation injury of many organs through various mechanisms, as shown in Figure 2. However, it is unclear whether MSC-EVs from multiple sources have different biological characteristics and treatment abilities.

### 4.1. Radiation-Induced Bone Injury

Ionizing radiation can induce bone damage in multiple ways. In terms of hematopoietic injury, ionizing radiation acts on hematopoietic stem cells, suppressing their proliferation and differentiation and reducing their number. Consequently, hematopoietic function is affected, and patients often present with pancytopenia, leading to anemia, bleeding, infection, and other symptoms. Moreover, the recovery of hematopoietic function is slow, and in severe cases, it can be life-threatening [53]. Regarding bone marrow damage, radiation reduces the number of bone marrow mesenchymal stem cells and increases apoptosis, thereby affecting their differentiation function. It also damages the vascular structure within the bone marrow and causes microcirculatory disorders. Eventually, bone marrow damage occurs, manifested as reduced bone marrow cell density, decreased hematopoietic tissue, increased fat tissue, and impaired normal physiological function of the bone marrow, resulting in decreased hematopoietic and immune functions [54]. In terms of bone loss, on the one hand, ionizing radiation inhibits the growth, survival, and functional maturation of osteoblasts, thereby reducing bone formation; on the other hand, it increases the number and activity of osteoclasts, thereby promoting bone resorption. This disrupts bone metabolic balance, leading to bone loss, reduced bone density, osteoporosis, decreased bone strength and toughness, increased bone fragility, and increased susceptibility to fractures. As for bone necrosis, high-dose radiation damages local blood circulation in bone tissue, causing vascular endothelial cell damage and vascular embolism, which lead to ischemia and hypoxia in bone tissue. Simultaneously, bone cells are damaged, and the metabolic and repair capacity of bone tissue is reduced, ultimately resulting in bone necrosis. It often occurs in areas such as the femoral head, causing pain and functional impairment in the affected area [55]. In severe cases, pathological fractures may occur, affecting patients’ mobility and quality of life. MSC-EVs demonstrate significant therapeutic potential in mitigating radiotherapy-induced bone injury by restoring the differentiation potential of BMSCs, alleviating oxidative stress, accelerating DNA repair, and promoting osteogenesis.

Researchers have evaluated the ability of BMSC-EVs to alleviate radiation-induced bone marrow stem cell injury in mice from 4 h to 7 days post-irradiation. Mice that were exposed to 500 cGy of radiation and intravenously injected with BMSC-EVs showed partial recovery of peripheral blood cell counts. In the mouse hematopoietic cell line FDC-P1, growth inhibition, DNA damage, and apoptosis caused by 500 cGy irradiation were all reversed by BMSC-EVs. This study suggests that MSC-EVs can reverse radiation-induced damage to bone marrow stem cells [56]. John et al. cultured BMSCs pretreated with lipopolysaccharide (LPS) on a large scale using hollow fiber bioreactors and isolated EVs via ultracentrifugation. Their study indicated that EVs from LPS-pretreated BMSCs can promote the recovery of hematopoietic function in irradiated mice and have potential for treating hematopoietic acute radiation syndrome. The RNA-seq results showed that the let-7 family and miR-143 were significantly upregulated in pretreated EVs. Additionally, monocytes treated with LPS-BMSC-EVs showed increased expression of genes such as IL-6, IDO, FGF-2, IL-7, IL-10, and IL-15. This suggests that LPS-BMSC-EVs can modulate immune cells and alter the immune environment [57]. Through complete blood counts and histological measurements, Kink et al. confirmed that LPS-pretreated BMSC-exos could affect macrophages, reduce radiation-injury clinical signs, and restore hematopoietic ability in bone marrow and spleen [58]. DPSC-EVs have also been demonstrated to reverse radiation-induced hematopoietic injuries [59].

BMSC-EVs could mitigate radiation-induced bone loss in rats that received 16 Gy of radiation to the left hind limb knee. At the cellular level, after co-incubating BMSC-EVs with BMSCs exposed to 6 Gy of radiation, the expression of γ-H2AX in BMSCs was significantly decreased. Moreover, BMSC-EVs could decrease radiation-induced increases in superoxide dismutase (SOD) 1 and 2, thereby enhancing cellular antioxidant effects [60]. This may be one mechanism by which BMSC-EVs improve bone loss in irradiated rats.

In summary, MSC-EVs exert therapeutic effects on radiation-induced bone injury by promoting the proliferation and differentiation of bone marrow mesenchymal stem cells, thereby increasing their numbers and enhancing bone repair. They modulate the immune microenvironment to suppress inflammation, aiding hematopoietic recovery and immune balance. Additionally, MSC-EVs stimulate angiogenesis to improve blood supply for bone repair and inhibit osteoclast activity while promoting osteoblast proliferation, thus increasing bone density and strength. Their mechanisms involve homing to injury sites, delivering bioactive molecules like growth factors and miRNAs, and regulating signaling pathways such as Wnt/β-catenin to control cell behavior and bone metabolism.

### 4.2. Radiation-Induced Lung Injury

Radiation-induced lung injury (RILI) includes early radiation pneumonitis and late radiation pulmonary fibrosis. The current treatment mainly relies on the application of high-dose glucocorticoids, which can provide short-term clinical relief. However, some patients continue to progress to irreversible pulmonary fibrosis. Lung transplantation is the most effective method for treating radiation-induced pulmonary fibrosis. Nevertheless, it is not commonly used due to the difficulty in matching donors and the wide range of post-transplant complications [61]. The pathogenic mechanism of RILI includes lung cell damage, vascular destruction, inflammatory response, oxidative stress, long-term fibrosis, and functional impairment. These mechanisms interweave with each other, ultimately leading to structural and functional damage to lung tissues [62].

Clinical and animal studies have reported miRNA changes in lung cancer patients post-radiotherapy. This implies miRNAs’ potential role in RILI’s pathological process [63]. MSC-EVs are rich in bioactive molecules like miRNAs, which may be a mechanism for treating RILI. Simultaneously, studies have shown that SARS-CoV-2-S-RBD-modified and miR-486-5p-engineered EVs derived from UC-MSCs could alleviate RILI [64]. This study developed engineered UC-MSC-EVs modified with both SARS-CoV-2-S-RBD and miR-486-5p, demonstrating that these engineered vesicles mitigate radiation-induced pulmonary injury by suppressing epithelial cell ferroptosis. Furthermore, miR-486-RBD-MSC-Exo inhibited radiation-induced ferroptosis and fibrosis in alveolar epithelial cells in vitro while ameliorating radiation-induced lung injury in mice in vivo. PL-MSC-EVs have been reported to mitigate radiation-induced pulmonary vascular injury, inflammation, and fibrosis through miRNA-214-3p, thereby alleviating RILI [65]. Li et al. found that BMSC-EVs treatment effectively alleviated RILI in mice, reduced collagen deposition, and lowered levels of IL-1β and IL-6 [66]. Meanwhile, in vitro results showed that BMSC-EVs treatment significantly reversed the radiation-induced EMT process in lung tissues. Among the enriched miRNA cargo in exosomes, miR-466f-3p mainly exerts a protective effect by inhibiting the AKT/GSK3β pathway. Mechanism studies further indicate that c-MET is a direct target of miR-466f-3p, and its recovery partially eliminates the inhibition of radiation-induced EMT process and AKT/GSK3 β signaling activity mediated by BMSC-EVs.

MSC-EVs are rich in bioactive molecules like miRNAs and proteins, which directly act on lung cells to regulate their behavior. For example, they deliver miRNAs (miR-486-5p, miR-466f-3p, and miRNA-214-3p) to modulate signaling pathways and suppress the expression of inflammation- and fibrosis-related genes.

### 4.3. Radiation-Induced Intestinal Injury

Regarding the intestine, ionizing radiation damages the intestinal mucosal barrier and hinders the self-renewal of epithelial cells. Small intestinal crypt stem cells are highly sensitive to radiation. Exposure to high doses leads to insufficient migration and differentiation of stem cells, disrupting the balance between proliferation and apoptosis, and thus inducing radiation-induced small intestinal injury [67].

Previous studies indicate that BMSC-EVs therapy boosts the proliferation of radiation-activated Lgr5^+^ cells while curbing apoptosis, thus effectively ameliorating radiation-induced colitis. The regulation of Lgr5^+^ cell proliferation and differentiation by BMSC-EVs is crucial in this process. This study indicates that BMSC-EVs have therapeutic effects on radiation enteritis and affect the proliferation and differentiation of Lgr5^+^ intestinal epithelial stem cells by regulating the Mir-195/Akt/β–catenin pathway [68]. Ning et al. showed that PL-MSC-EVs targeted radiation-damaged intestines in a dose-dependent manner. By using molecular imaging to quantify PL-MSC-EVs in damaged intestines, this study offers spatiotemporal information for early injury diagnosis. Also, through reducing apoptosis, promoting angiogenesis, and improving the intestinal inflammatory environment, PL-MSC-EVs exhibit superior nano-functions. Moreover, miRNA-455-5p in PL-MSC-EVs negatively regulates SOCS3 expression, and the activated downstream Stat3 signaling pathway is involved in the therapeutic effects of PL-MSC-EVs on radiation-induced intestinal injury [69]. This helps guide personalized treatment and provides data for designing EVs-based theranostic strategies to promote recovery from radiation-induced intestinal injury and offers a cell-free treatment option for radiotherapy.

MSC-EVs ameliorate radiation-induced colitis by suppressing inflammation and promoting intestinal epithelial repair. They enhance the expression of genes related to gut barrier function, thereby maintaining epithelial integrity. Additionally, they boost the proliferation and inhibit the apoptosis of Lgr5+ intestinal stem cells. Mechanistically, MSC-EVs achieve these effects by delivering miRNA-455-5p, which downregulates SOCS3 and activates the Stat3 pathway.

### 4.4. Radiation-Induced Skin Injury

Skin, being the largest human organ, is prone to radiation damage due to the high proliferation rate of epithelial cells. Radiation-induced skin damage can be categorized into early effects (acute damage) and late effects (chronic damage). Early effects include erythema, hyperpigmentation, hair loss, dry desquamation, wet desquamation, and ulceration, while late effects involve late-onset ulcers, ischemia, fibrosis, atrophy, and malignant skin changes [70,71].

A study showed that ADSC-EVs could significantly promote the healing of radiation-induced skin damage in an established in vitro model using human dermal fibroblasts [72]. It was found that ADSC-EVs enhanced skin regeneration by promoting human dermal fibroblast proliferation and migration while inhibiting differentiation, thereby accelerating wound healing, increasing epidermal regeneration and dermal repair at the damaged skin site, shortening healing time, and reducing scar formation. Moreover, ADSC-EVs, with immune regulatory functions, inhibited the expression of inflammation-related genes, which reduced inflammatory exudation and tissue damage and created a favorable microenvironment for skin repair. Furthermore, ADSC-EVs significantly promoted the expression of collagen-related genes, which eventually increased collagen synthesis and improved radiation-induced skin recovery. These findings suggest that ADSC-EVs treatment can suppress inflammation and promote extracellular matrix deposition.

### 4.5. Radiation-Induced Brain Injury

Radiation-induced brain injury primarily affects neurons, glial cells (including astrocytes, oligodendrocytes, and microglia), vascular endothelial cells, and neural stem cells. Neuronal damage and death, particularly in neurogenic regions such as the hippocampus, can impair learning and memory. The activation and apoptosis of glial cells lead to neuroinflammation and tissue damage. Damage to vascular endothelial cells disrupts the blood–brain barrier, thereby triggering neuroinflammation. Injury and death of neural stem cells inhibit neurogenesis and are associated with cognitive impairment. These cellular injuries collectively contribute to the pathological processes of radiation-induced brain injury, including neuroinflammation, neurodegeneration, and cognitive dysfunction [73].

Previous experimental studies have revealed that ADSC-EVs can significantly improve both the viability and clonogenic activity of neural stem cells following 1 Gy radiation exposure [74], potentially representing a key mechanism through which ADSC-EVs protect neural stem cells from radiation-induced injury. Liu et al. reported that ADSC-EVs exert protective effects in rats exposed to 30 Gy cranial irradiation, potentially through SIRT1 pathway activation, which reduces oxidative stress, inflammation, and microglial infiltration in irradiated brain tissues [75].

### 4.6. Radiation-Induced Cardiac Injury

Radiation-induced cardiac injuries encompass a variety of conditions, including myocarditis, pericarditis, coronary artery disease, arrhythmia, and valvular heart disease. However, clinical treatment options for these conditions are currently limited [76].

UC-MSC-EVs inherit the low immunogenicity and immunomodulatory properties of their parent UC-MSCs. Hu’s study demonstrated that UC-MSC-EVs could mitigate radiation-induced structural cardiac injury. In their experiment, mice received localized 20 Gy X-ray irradiation to the heart, followed by intravenous administration of UC-MSC-EVs via tail vein injection the next day. Post-experimental observations revealed that control group mice exhibited disrupted mitochondrial structure and decreased mitochondrial membrane potential in cardiac tissues compared to the treatment group. They also found that UC-MSC-EVs protected against radiation-induced cardiac organoid injury through modulation of oxidative phosphorylation and p53 signaling pathways [77]. Luo et al. isolated PL-MSC-EVs following 5 Gy γ-ray irradiation and found no differences in EV quantity or size distribution between irradiated and non-irradiated MSCs; however, significant miRNA sequencing variations were observed. In vitro experiments using human umbilical vein endothelial cells (HUVECs) and H9c2 cells demonstrated that post-irradiation PL-MSC-EVs significantly reduced cell proliferation while increasing apoptosis and DNA injury. This study suggests that PL-MSC-EVs may potentially induce radiation-induced cardiotoxicity and exhibit adverse effects [78].

### 4.7. Other Radiation-Induced Injuries

Radiation-induced liver injury is a complex pathological process characterized by damage to hepatocytes, cholangiocytes, and endothelial cells, as well as the induction of fibrosis and inflammation. This causes sinusoidal endothelial cell toxicity, hepatocyte atrophy, and veno-occlusive disease, which gradually lead to loss of liver function and progression to radiation-induced liver disease (RILD) [79]. EVs injected via the tail vein tend to accumulate in the liver due to several factors. The liver’s rich blood supply and unique microenvironment, featuring cells like Kupffer cells and hepatic sinusoidal endothelial cells, facilitate the filtration and uptake of blood-borne substances, including exosomes [80]. The liver’s reticuloendothelial system, particularly Kupffer cells, actively phagocytoses particulate matter such as exosomes. Additionally, the physical and chemical properties of EVs, such as surface proteins and lipids, may interact with liver cell receptors, resulting in enhanced accumulation. Research on MSC-EVs therapy for radiation-induced liver injury is limited, and more experimental data are needed to confirm its efficacy. Although studies have shown that MSC-EVs can protect against other types of liver injury, there is little direct evidence for their effectiveness against radiation-induced liver damage.

For the radiation-induced bladder injuries, also named radiation-induced cystitis, BMSC-EVs were proven to have therapeutic effects [81]. Human bladder fibroblasts (HUBFs) were irradiated at 3 Gy and then co-cultured with BMSC-EVs. It was found that MSC-EVs downregulated αSMA and CTGF transcription, induced anti-fibrotic cytokine secretion (IFN, IL-10, and IL-27), and decreased secretion of pro-fibrotic cytokines (IGFBP2, IL-1, IL-6, IL-18, PDGF, TNF, and HGF), which finally inhibited fibrosis and inflammation. Moreover, BMSC-EVs could also induce vessels from HUVEC cells and had a proangiogenic role in radiation-induced cystitis. In conclusion, BMSC-EVs have a promising therapeutic potential in radiation-induced cystitis by inhibiting fibrosis, inflammation, and vascular damage.

UC-MSC-EVs could ameliorate uranium-induced renal injury by improving kidney and bone marrow morphology while enhancing uranium excretion and reducing its body retention, with protective effects surpassing those of UC-MSCs themselves. Proteomic analysis reveals that UC-MSC-EVs competitively bind uranyl ions through transferrin and other proteins, thereby reducing their deposition in renal tubular cells. They regulate glutathione metabolism and related pathways to enhance antioxidant capacity, thereby decreasing mitochondrial ROS production and alleviating lipid peroxidation, which collectively reduce cellular apoptosis and ferroptosis [82].

In a chemotherapy-induced premature ovarian insufficiency (POI) rat model, intravenous administration of UC-MSC-EVs significantly improved ovarian function. UC-MSC-EVs modulated the local ovarian microenvironment through involvement in key signaling pathways related to immunoregulation, cell viability, inflammatory modulation, fibrosis, and metabolism, thereby preserving fertility in POI rats. This study indicated that UC-MSC-EVs may have the same effects on radiation-induced ovarian injury [83].

Furthermore, DP-MSC-EVs play a significant role in radiation-induced senescence. In this study, a model of submandibular gland radiation injury was established by exposing 7-week-old mice to 25 Gy of radiation. Treatment with DP-MSC-EVs significantly reduced the expression of inflammatory cytokines and aging-related genes in the mice. Additionally, DP-MSC-EVs ameliorated oxidative stress in submandibular gland cells, thereby inhibiting radiation-induced cellular senescence [84].

Radiotherapy is crucial for the treatment of head and neck cancers, yet it can damage surrounding tissues like salivary glands, leading to degeneration, reduced vascularity, and fibrosis, and thus causing irreversible damage and xerostomia. For the radiation-induced salivary gland injury, ADSC-EVs were shown to mitigate the injury and improve salivary function [85]. In this study, a 14 Gy irradiated salivary gland animal model was established, and the researchers found the treatment mechanism was that ADSC-EVs delivered miR-199a-3p to target Twist1 and subsequently modulate the TGF-1β/Smad3 signaling pathway, which effectively attenuated epithelial–mesenchymal transition (EMT) and inhibited fibrosis.

The therapeutic effects of MSC-EVs from different sources in different radiation-injured organs are summarized from the published literature and listed in Table 1.

## 5. Clinical Development, Potential Limitations, and Challenges of EVs

### 5.1. Clinical Development of EVs

Currently, there are no EVs specifically approved for clinical use against radiation damage. However, some clinical trials are exploring related applications in which the EVs were used as biomarkers for liquid biopsy to diagnose diseases and as therapeutic tools.

EVs have stable properties and can be found in all bodily fluids, while specific components such as nucleic acids and proteins change when tumor cells secrete EVs, thereby serving as biomarkers for tumor diagnosis [86]. The expression level of miR-21 in serum exosomes of esophageal cancer patients is correlated with tumor staging, lymph node involvement, and tumor metastasis [87]. In patients with pancreatic cancer, the cell membrane anchor protein GPC1 is specifically overexpressed in the blood vesicles; its sensitivity and specificity in the diagnosis of pancreatic cancer are 100%, and the elevation of GPC1 in the blood vesicles is earlier than the imaging changes [88]. Most clinical trials on EVs involve the use of vesicles in the diagnosis and prognosis of several types of cancer (liquid biopsy). However, in most studies, EVs are examined as “secondary” rather than disease-specific markers.

MSC-EVs were used as therapeutic tools in two clinical studies. Shi et al. investigated the biological distribution and effects of nebulized inhalation of human adipose-derived MSC-EVs (haMSC-EVs) in preclinical lung injury models and proved the safety of nebulized inhalation of haMSC-EVs in healthy volunteers, providing support for the clinical translation of EVs [89]. In another intervention study, the safety and effectiveness of MSC-derived exosomes as aerosol inhalation was explored in the treatment of novel coronavirus pneumonia (NCP). This study recruited thirty participants and divided them into three groups: two of them received standard treatment while inhaling 3 mL of a solution containing 0.5–2 × 10^10^ EVs and the placebo group received standard treatment, inhaling 3 mL of nanoparticle-free solution twice a day. All patients enrolled in this study did not experience any adverse events throughout the entire trial and inhalation process. Furthermore, patients treated with extracellular vesicles showed lower levels of C-reactive protein compared to the placebo group at the end of treatment, indicating a reduction in inflammatory processes after EV therapy [90].

At the preclinical and clinical levels, some technical and functional deficiencies still need to be addressed. More detailed molecular spectrum analysis of vesicular cargo and stricter characterization of EVs should be conducted to further determine the minimum content for specific applications. Unfortunately, most of the selected clinical trials involving the use of EVs have not reported research results. Nevertheless, the great interest and extensive work in the development of EVs indicate their widespread clinical application.

### 5.2. Potential Limitations and Challenges in the Clinical Translation of EVs

MSC-EVs hold great promise for the treatment of radiation injuries due to their regenerative and anti-inflammatory properties. However, several challenges and limitations hinder MSC-EVs’ clinical translation, including large-scale production and isolation, standardization and characterization, pharmacokinetics and targeting, safety profile, dose optimization, and administration route.

MSC-EVs are a promising cell-free therapy for clinical applications. However, the commonly used methods for obtaining MSC-EVs, such as density gradient centrifugation, differential centrifugation, and commercial assay kits, have several limitations. These methods often result in low yield and purity, are time-consuming, and require expensive equipment [91,92]. Additionally, they may introduce contaminants such as soluble proteins and nucleic acids, which can affect the functional properties of the isolated EVs. Furthermore, the high centrifugal forces used in these methods may alter the structural integrity of the EVs, potentially impacting their therapeutic efficacy.

The heterogeneity of MSC-EVs, which is influenced by factors such as cell source, culture conditions, and isolation methods, presents significant challenges to standardization and quality control. For instance, MSCs can be derived from various tissues, including bone marrow, adipose tissue, and umbilical cord, each potentially yielding EVs with different compositions and functionalities. This heterogeneity severely restricts the quality control and management of MSC-EVs as drugs and increases the problem of drug resistance, resulting in limited reproducibility of functional measurements in vitro and in vivo [93]. Additionally, the lack of reliable tools and specific markers to distinguish EV subtypes further complicates their classification and functional assessment.

Current research indicates that MSC-EVs can aggregate at injury sites, akin to their parental cells, but the precise mechanisms underlying this targeting remain elusive. For instance, MFGE8, a lipophilic glycoprotein enriched in MSC-EVs, has been shown to bind to phosphatidylserine (PS) on injured cells, facilitating the targeting of EVs to damaged tissues and the subsequent release of therapeutic factors [94]. However, the complexity of EV components, including various miRNAs and proteins, suggests that multiple pathways may be involved in their therapeutic effects. While some miRNAs have been implicated in the repair process, the roles of other components remain to be fully elucidated [95].

The natural targeting properties of MSC-EVs, their extended retention time at injury sites, and their ability to bypass certain biological barriers make them a compelling option for targeted therapy. However, the majority of them are preferentially recruited to the liver and lung in vivo. The liver, as the central hub of human metabolism, possesses robust phagocytic and clearance capabilities. Hepatic sinusoidal endothelial cells and Kupffer cells play pivotal roles in effectively eliminating xenobiotics from the circulation, with EVs being no exception [96,97]. The lungs, critical organs for gas exchange, boast an extensive capillary network and a multitude of alveolar epithelial cells. Upon entering the circulatory system, EVs traverse the pulmonary circulation with blood flow [98]. Owing to the relatively small diameter of pulmonary capillaries, EVs tend to be retained in this location. Concurrently, alveolar epithelial cells and alveolar macrophages exhibit vigorous phagocytic and uptake activities, enabling them to clear inhaled particulate matter and EVs, thereby contributing to the accumulation of EVs in lungs. Despite these insights, further research is needed to comprehensively understand the pharmacokinetics and targeting mechanisms of MSC-EVs, as well as to develop strategies for enhancing their tissue-specific delivery and therapeutic efficacy in the context of radiation injury repair.

Despite their advantages, including low immunogenicity and the inability to self-replicate, which reduces the risk of tumor formation, the safety profile of MSC-EVs still requires comprehensive evaluation [99]. However, potential toxicities from repeated administration and the long-term effects of engineered MSC-EVs remain areas of concern [100]. While MSC-EVs are generally considered safer than cell-based therapies, the lack of standardized protocols and detailed toxicity evaluation tools poses challenges. Further research is necessary to thoroughly assess the safety profile of MSC-EVs, particularly focusing on their long-term effects and potential toxicities, thereby ensuring their safe and effective application in clinical settings.

Current research highlights the complexity of determining the ideal dose, as it depends on factors such as the disease model, EV composition, and administration method. For instance, studies have shown that the therapeutic effects of MSC-EVs can vary significantly with different dosages, with some models demonstrating a dose–response relationship, while others reveal diminishing returns at higher doses [101]. Further preclinical studies are needed to systematically evaluate the impact of dosage and administration route on therapeutic outcomes, ensuring the safe and effective translation of MSC-EV-based therapies into clinical practice. MSC-EVs primarily work by regulating endogenous stem cells to regenerate damaged tissue. After low-dose radiation, donor cells partially promote tissue regeneration and partly stimulate endogenous stem cells. However, if a large radiation dose depletes most endogenous stem cells in the irradiated area, regeneration will nearly entirely depend on transplanted stem cells. Therefore, after high-dose radiation, MSC-EVs may not be effective enough, and a larger dose of stem cells will be needed [102].

Addressing these challenges through improved production techniques, standardized protocols, and rigorous preclinical studies is essential to advance MSC-EV-based therapies for radiation injury repair into clinical practice.

## 6. Conclusions

Existing data suggest that MSC-EVs exhibit considerable therapeutic potential in radiation-induced injury of different organs by reducing oxidative stress, reversing EMT, exerting anti-inflammatory effects, and decreasing stem cell apoptosis. However, the development of MSC-EV-based radiation injury therapy drugs still faces challenges, such as the need for standardized MSC-EV extraction, characterization methods, optimized dosing regimens and routes, and a deeper understanding of the molecular mechanisms.

## Figures and Tables

**Figure 1 pharmaceutics-17-00652-f001:**
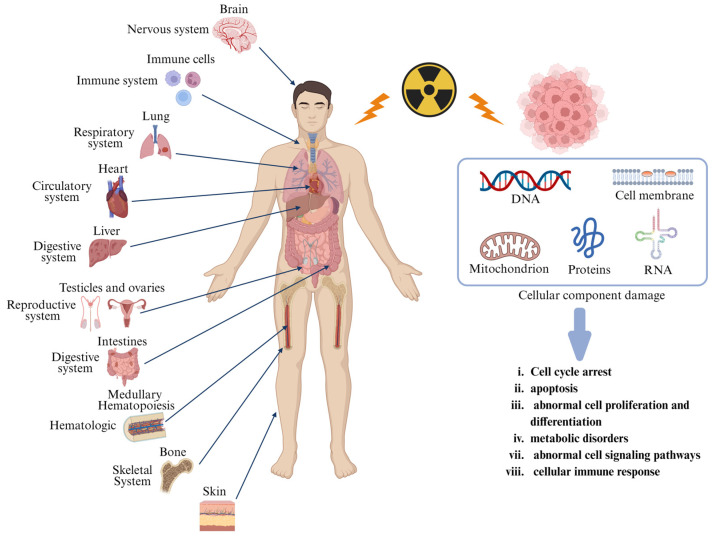
Radiation causes damage at the level of organs and cells. Radiation can damage multiple organs and systems in the human body, including the bone marrow, intestines, lungs, liver, brain, reproductive system, and skin. Radiation damage to cells is multifaceted, involving various cellular components such as DNA, cell membranes, mitochondria, proteins, RNA, etc.

**Figure 2 pharmaceutics-17-00652-f002:**
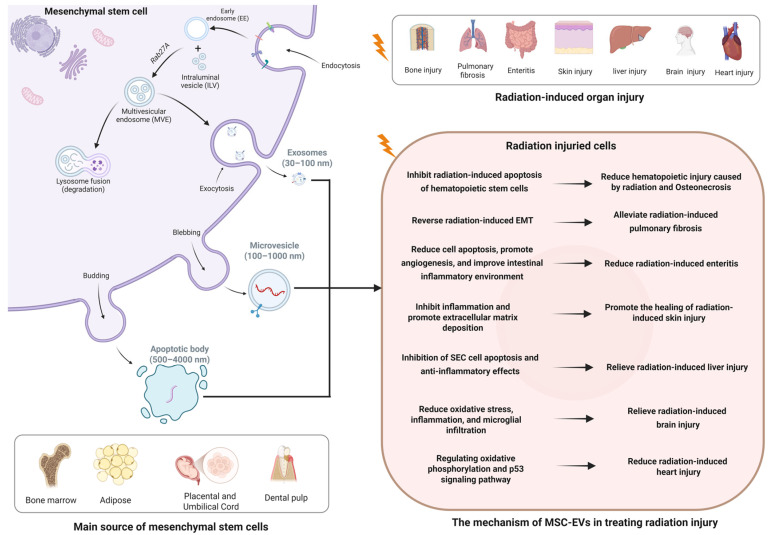
The therapeutic effects and mechanisms of MSC-EVs from different sources in radiation injury. MSCs secrete vesicles through exocytosis, sprouting, and foaming. MSC-EVs treat radiation injury by reducing oxidative stress, reversing EMT, exerting anti-inflammatory effects, and reducing stem cell apoptosis. EMT: epithelial–mesenchymal transition; SEC: sinus endothelial cells.

**Table 1 pharmaceutics-17-00652-t001:** Summary of the therapeutic effects of MSC-EVs in different radiation-induced injuries.

Radiation Injuries	Source of EVs	Ingredients	References
Radiation-induced hematopoietic injury	BMSC-EVs DPSC-EVs	Let-7 miR-143	[56,57,58,59]
Radiation-induced bone loss	BMSC-EVs	--	[60]
Radiation-induced lung injury	UCMSC-EVs PL-MSC-EVs BMSC-EVs	miR-486-5p miR-214-3p MiR-466f-3p	[64,65,66]
Radiation-induced intestinal injury	PL-MSC-EVs BMSC-EVs	miR-195 miR-455-3p	[68,69]
Radiation-induced skin injury	ADSC-EVs	--	[72]
Radiation-induced brain injury	ADSC-EVs	--	[75]
Radiation-induced heart disease	PL-MSC-EVs UCMSC-EVs	miR-23a-5p miR-29a-3p miR-146a-5p	[77]
Radiation-induced cystitis	BMSC-EVs	--	[81]
Radiation-induced kidney injury	UC-MSC-EVs	--	[82]
Radiation-induced premature ovarian injury	UC-MSC-EVs	--	[83]
Radiation-induced salivary gland injury	ADSC-EVs	miR-199a-3p	[85]

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
