# Peer review of "From Mechanism to Therapy: The Role of MSC-EVs in Alleviating Radiation-Induced Injuries"

_pharmaceutics, 2025, doi:10.3390/pharmaceutics17050652_

Round 1
Reviewer 1 Report
Comments and Suggestions for Authors
Through this review article, C.Huang et al described recent advances to utilize MSC-EVs for the treatment of various radiation-induced injuries.
The authors firstly described the pathophysiology of radiation injury, before providing overview about MSC-derived EVs’ properties, and their recent applications categorized based on the source of MSC cells. Finally, the authors provide their views on future prospect.
Overall, the review is of appropriate depth and length, informing readers with notable examples of MSC-EV development in relation with Radiation injury. There are few aspects which require further clarifications and revisions however, before I can suggest the acceptance of this article to Pharmaceutics:
- In subsection 2, the authors put Current Therapeutic Landscape of Radiation Injury as the title. However, the text description focuses only on pathophysiology of radiation injury, without touching current therapy or those in development for this. Please amend and include this information.
- Why did the authors choose to categorize the subsection based on origin of MSC cells? Are there specific characteristics of the EVs derived from these MSCs? Please elaborate
- In relation with above comment, it would be good if the authors can provide a summary table of their characteristics and how it relates to certain applications they are utilized for.
- Are there any EV-based therapeutics that are in clinical development for radiation injury or similar types of injury? Please describe them separately as compared to those in preclinical stage evaluation.
Author Response
Comments 1: In subsection 2, the authors put Current Therapeutic Landscape of Radiation Injury as the title. However, the text description focuses only on pathophysiology of radiation injury, without touching current therapy or those in development for this. Please amend and include this information.
Response 1: Thank you for pointing this out. We agree with this comment. In the second part of the revised manuscript, we added information about current or developing treatment methods for various radiation injury diseases. The added content has been marked in red font in the revised manuscript (Pages 5-8, lines 121-201). The treatment strategies for radiation-induced injury diseases cover various means, including drug therapy, surgical treatment, hyperbaric oxygen therapy, rehabilitation therapy, stem cell therapy, etc. However, there are many limitations, such as short-term or uncertain drug efficacy, large surgical trauma and high risk, lack of large-scale experimental support for hyperbaric oxygen therapy, and immature stem cell technology.
Comments 2: Why did the authors choose to categorize the subsection based on origin of MSC cells? Are there specific characteristics of the EVs derived from these MSCs? Please elaborate.
In relation with above comment, it would be good if the authors can provide a summary table of their characteristics and how it relates to certain applications they are utilized for.
Response 2: Thank you for pointing this out. We agree with this comment. MSCs from different organizational sources mainly differ in cell acquisition and immune response. Different resources give the MSCs unique features, for instance, BMSC is the firstly discovered and widely used MSCs; ADSC and DPSC are easier and safer to obtain; PL-MSC and UCMSC have stronger clone formation and differentiation ability. According to previous literatures, the difference in the physical and biological properties of EVs secreted by MSCs from different tissue sources is not clear, but the different contents they contain may play a key role in regulating target cells.
When different organs are subjected to radiation damage, they exhibit tissue-specific damage patterns. To better demonstrate the therapeutic characteristics of MSC EVs for different radiation injury diseases, we have changed the categories of the subsection into different radiation-induced organ injuries. A clearer display of the specific characteristics of different radiation injury diseases, as well as the therapeutic effects and mechanisms of MSC-EVs from different sources in various radiation injury diseases, can enhance logical coherence.
The fourth part of the manuscript has been revised to discuss the therapeutic role of MSC-EVs in various radiation injury diseases, with subheadings as follows: 4.1 Radiation-induced bone injury; 4.2 Radiation-induced lung injury; 4.3 Radiation-induced intestinal injury; 4.4 Radiation-induced skin injury; 4.5 Radiation-induced brain injury; 4.6 Radiation-induced cardiac injury; 4.7 Other radiation-induced injuries. The manuscript has been revised on pages 9-19, lines 245-516. The revised content has been marked in red font.
We also have added Table 1 (pages19,pages 519-520) regarding the therapeutic effects of MSC-EVs in various radiation injuries to facilitate readers' understanding of the roles of EVs from different sources in different radiation injury diseases.
Comments 3:Are there any EV-based therapeutics that are in clinical development for radiation injury or similar types of injury? Please describe them separately as compared to those in preclinical stage evaluation.
Response 3: Thank you for pointing this out. We agree with this comment. Currently, there are no EVs specifically approved for clinical use against radiation damage. However, some clinical trials are exploring related applications. EVs are ubiquitous in intercellular communication and can be detected in tissues and body fluids. Their complex cargo reflects the (pathological) physiological state of the cells from which they originate. EVs, as biomarkers for disease diagnosis in clinical practice, have been applied in various diseases. However, EVs are currently still used as an auxiliary diagnostic tool, and clinical diagnosis is not solely based on the biomarkers of EVs to determine diseases. EVs have undergone clinical trials in various diseases, but unfortunately the results of the trials are not comprehensive. The specific details of the clinical trials have been described in detail in the revised manuscript (Pages 20-21, paragraphs 1-5,lines 522-559).

Reviewer 2 Report
Comments and Suggestions for Authors
The manuscript provides a comprehensive overview of the importance of using EVs as a strategy to counteract radiation-induced injuries. It would be important to describe the process by which EVs are used to fully understand it.
Author Response
Comments 1:The manuscript provides a comprehensive overview of the importance of using EVs as a strategy to counteract radiation-induced injuries. It would be important to describe the process by which EVs are used to fully understand it.
Response 1:
Thank you for pointing this out. We agree with this comment. Thank you very much for your time and effort in reviewing our manuscript. We would like to express our gratitude to you for your invaluable and constructive comments to help us significantly improve the manuscript. The revisions we have made in response to the reviewer's comments will be highlighted in red in the revised manuscript.
We have rechecked the manuscript and found some sentences with imprecise descriptions, which have been revised and marked in red font. The fourth part of the manuscript has been revised to discuss the therapeutic role of MSC EVs in various radiation injury diseases, with subheadings as follows: 4.1 Radiation-induced bone injury; 4.2 Radiation-induced lung injury; 4.3 Radiation-induced intestinal injury; 4.4 Radiation-induced skin injury; 4.5 Radiation-induced brain injury; 4.6 Radiation-induced cardiac injury; 4.7 Other radiation-induced injuries (pages 9-19, lines 245-516). Providing a clearer display of the specific characteristics of different radiation injury diseases, as well as the therapeutic effects and mechanisms of MSC EVs from different sources in various radiation injury diseases, can enhance logical coherence. We also have added Table 1 (pages19,lines 519-520) regarding the therapeutic effects of MSC-EVs in various radiation injuries to facilitate the reader’s understanding of the roles of EVs from different sources in different radiation injury diseases. We also have added a new section -- section 5 “Clinical development, potential limitations and challenges of MSC-EVs” in the revised manuscript, to discuss the limitations of clinical application of MSC EVs.

Reviewer 3 Report
Comments and Suggestions for Authors
Recently, MSC -EV have drawn a lot of attention due to potential regenerative capabilities from injuries. In the current review, Huang et al, have compiled literature EVs derived from Bone Marrow, Dental pulp, Adipose-derived stem cells, Umbilical cord mesenchymal stem cells in context to alleviate Radiation-Induced Injury. Authors have addressed MSC-EVs potential positive therapeutic effect, mechanisms and future challenges. Overall, the review manuscript flows very well. Similar articles, PMID: 39780320PMID: 40135580; PMID: 39679884 shares similar vision for MSC-EV and has not been cited. Here are some suggestions to increase basic understanding of MSC-EV.
- All cells secrete EVs, authors should specify unique characteristics of MSC-EV which are different than normal EV. Are bone marrow, adipose, dental, umbilical chod cells derived EV are similar?
- miRNAs, proteins. ..etc., have been often referred to EV contents, it will be better if author can make a table which distinguish between them.
- EVs are very heterogenous due to their origin, cell specificity, presence of surface and internal contents of coding/non-coding RNAs which have biological functions which are unknown. Authors should discuss caution how MSC-EV can be used as therapeutic without knowing their action of mechanism.
- EVs field have challenges due to isolation and characterization methods, Authors should also discuss these types of challenges in the field of MSC-EV and some future prospective.
Author Response
Comments 1:All cells secrete EVs, authors should specify unique characteristics of MSC-EV which are different than normal EV. Are bone marrow, adipose, dental, umbilical chod cells derived EV are similar?
Response 1:Thank you for pointing this out. We agree with this comment. EVs derived from bone marrow, adipose, dental pulp, and umbilical cord cells have similarities in basic composition, immune regulation, promotion of tissue repair and regeneration, and low immunogenicity, but they also differ in certain specific functions and components. However, the literatures of compare in MSC-EVs from different tissue origins were limited. Providing a clearer display of the specific characteristics of different radiation injury diseases in different organs, as well as the therapeutic effects and mechanisms of MSC-EVs from different sources in various radiation injury diseases, can enhance logical coherence. So, we revised the fourth part of the manuscript to discuss the therapeutic role of MSC-EVs in various radiation injury diseases, with subheadings as follows: 4.1 Radiation-induced bone injury; 4.2 Radiation-induced lung injury; 4.3 Radiation-induced intestinal injury; 4.4 Radiation-induced skin injury; 4.5 Radiation-induced brain injury; 4.6 Radiation-induced cardiac injury; 4.7 Other radiation-induced injuries. The above modifications have been marked in red font in the document (pages 9-19, lines 245-516).
Comments 2: miRNAs, proteins. ..etc., have been often referred to EV contents, it will be better if author can make a table which distinguish between them.
Response 2: Thank you for pointing this out. We agree with this comment. We have added Table 1 (pages19,pages 519-520) regarding the therapeutic effects of MSC-EVs in various radiation injuries to facilitate readers' understanding of the roles of EVs from different sources in different radiation injury diseases.
Comments 3: EVs are very heterogenous due to their origin, cell specificity, presence of surface and internal contents of coding/non-coding RNAs which have biological functions which are unknown. Authors should discuss caution how MSC-EV can be used as therapeutic without knowing their action of mechanism.
Response 3: Thank you for pointing this out. We agree with this comment. The heterogeneity of MSC-EVs, which is influenced by factors such as cell source, culture conditions, and isolation methods, presents significant challenges to standardization and quality control. For instance, MSCs can be derived from various tissues including bone marrow, adipose tissue, and umbilical cord, each potentially yielding EVs with different compositions and functionalities. This heterogeneity severely restricts the quality control and management of MSCs and their EVs as drugs, and increases the problem of drug resistance, resulting in limited reproducibility of functional measurements in vitro and in vivo. Additionally, the lack of reliable tools and specific markers to distinguish EVs subtypes further complicates their classification and functional assessment. We added discussion about these issues in the revised manuscript (pages 21-22, lines575-584).
Comments 4: EVs field have challenges due to isolation and characterization methods, Authors should also discuss these types of challenges in the field of MSC-EV and some future prospective.
Response 4: Thank you for pointing this out. We agree with this comment. Although MSC-EVs have shown positive therapeutic effects in various clinical diseases, there are still potential limitations and challenges in their clinical application. Therefore, we have added a new section -- section 5 “Clinical development, potential limitations and challenges of MSC-EVs” in the revised manuscript, to discuss the limitations of clinical application of MSC EVs (pages 21-23, paragraph 1-8, lines 560-637). We have discussed in detail from the following aspects: 1. difficulties in large-scale production, 2. imperfect purification and separation technology, 3. difficulty in quality control, 4. insufficient research on content and mechanism, 5. potential toxicity concerns, 6. immunogenicity issues, etc. In addition, as a paracrine pathway of MSCs, if the target cells are severely damaged during radiation injury, the regulatory effect of MSC EVs on the target cells will be very limited.
